# Facilitators and barriers for healthcare workers' adherence to the national nutritional guidelines for people living with HIV in Dar-es-Salaam: A mixed-method study

**Felistar Mwakasungura**[1], **Rebecca Mkumbwa**[1]*, **Bruno Sunguya**[2]

**1** Department of Development Studies, School of Public Health and Social Sciences, Muhimbili University of Health and Allied Sciences, Dar es Salaam, Tanzania, **2** Department of Community Health, School of Public Health and Social Sciences, Muhimbili University of Health and Allied Sciences, Dar es Salaam, Tanzania

* dicksonrebecca6@gmail.com

## Abstract

The dual burden of HIV and malnutrition is a significant public health challenge in high burden Sub-Saharan African countries. Tanzania is no exception. Care and treatment of HIV has been strengthened in these countries, however integration with nutrition care and support remains poor. This study evaluated adherence to nutritional guidelines for people living with HIV (PLHIV) in Dar es Salaam Tanzania and identified its influencing factors. A mixed method study was conducted among PLHIV, health facility administrators, and healthcare providers from April 29 to May 27, 2024. Data were collected through observation checklists, structured questionnaires, and in-depth interviews. Statistical and qualitative analyses were performed to assess adherence and determinants. Among 478 participants who received care, only 19.5% were managed while adhering to the national nutritional guidelines. Universally, anthropometric assessments were taken, however, only 1% received micronutrient supplements. Higher education (AOR = 5.08, p = 0.019) and attendance at referral hospitals (AOR = 8.23, p = 0.032) positively influenced adherence. Conversely, nurse attendance (AOR = 0.34, p = 0.038), adequate staffing (AOR = 0.31, p = 0.010), and urban residence (AOR = 0.47, p = 0.009) negatively influenced adherence. Key facilitators included consistent training and supportive leadership while financial constraints and high staff turnover remained important barriers. There is a significant gap in adherence to nutritional guidelines among PLHIV in Dar es Salaam, Tanzania. This highlights the need for improved resource distribution, staff training, and clients education in Tanzania and countries with similar contexts.

## Introduction

Human Immunodeficiency Virus (HIV) remains a critical global health challenge, with approximately 39.9 million people living with the virus as of 2023 [1]. Characterized by immune system suppression, HIV exacerbates nutritional challenges [2], particularly in low- and middle-income countries (LMICs). Sub-Saharan Africa which accounts for 65% of the

**Data availability statement:** This paper contains extracts from the qualitative data gathered and analyzed. Publicly sharing the raw data would contradict the approved ethical application and the information provided to participants regarding the study and the use of their data. The Institutional Review Board (IRB) requires that a formal data sharing agreement should be in place before releasing data. Therefore, upon request and following the stipulated Tanzanian policies on data sharing, data may be made available to researchers requesting access. Requests can be directed to Muhimbili University of Health and Allied Sciences, Institutional Review Board through drpi@muhas.ac.tz

**Funding:** The authors received no specific funding for this work.

**Competing interests:** The authors have declared that no competing interests exist.

global HIV population [1], harbor 23.7% of malnutrition burden among People Living with HIV (PLHIV) [3]. This is comparable to the regional burden of malnutrition in the region [4]. In Tanzania, over 1.5 million individuals were living with HIV by the year 2023, and approximately 4.7% of these individuals reside in Dar es Salaam [5], a business capital with 10% of the national population [6].

The interaction between HIV and malnutrition creates a detrimental cycle [7]. HIV not only increases nutritional requirements [8], but also impairs the body's ability to meet these needs due to symptoms such as oral thrush, painful swallowing, and nausea [9]. As a result, asymptomatic HIV-positive individuals need 10% more energy, while those with symptomatic HIV require 20–30% more energy compared to HIV-negative individuals [10]. This heightens energy demand coupled with reduced food intake further compromises the already weakened immune system [8]. Nutritional care, when integrated with antiretroviral therapy (ART), can significantly improve health outcomes [11]. However, despite ART's success in reducing HIV-related morbidity and mortality, nutritional concerns persist [12]. The rolled-out care and treatment for HIV in the region and countries with high burden of HIV has not been in parallel with efforts to integrate nutritional care services for PLHIV. This is mainly due to a human resources for health challenges, particularly those with competences to identify, manage, and provide counseling and treatment of various nutritional challenges [13]. As a result, one in ten PLHIV in attending Care and Treatment Centers (CTCs) for HIV are undernourished while others have overweight or obesity [11].

Tanzania has the HIV prevalence of 4.9% among its adult population [14]. The country has made significant strides in reaching the national and global HIV response targets, In line with the 95-95-95 targets [5]. About 83% of adults living with HIV are aware of their HIV-positive status [5]. Among those aware, 97.9% are on ART, 94.3% have achieved viral load suppression [5]. Despite its widespread ART services up to the primary health care level, it has a burden of 19.4% of PLHIV attending these CTCs malnourished [10].

The Tanzanian government has remained committed to addressing both HIV and malnutrition through policies and guidelines. The *"National Guidelines for Nutrition Care and Support for People Living with HIV"* and the *"Health Sector HIV/AIDS Strategic Plan"* [7] are some of the deliberate policy efforts in addressing the burden. The frameworks aim to improve nutritional care by integrating it into comprehensive health services, including assessment, education, counseling, and supplementation [7]. The *"National HIV/AIDS Council Strategic Framework 2011–2015"* emphasizes the importance of healthy eating and lifestyle as part of HIV care. The 2016 national guidelines for Nutrition Care and Support aim to coordinate nutrition programming, provide consistent information, and ensure that healthcare services include essential nutrition components such as assessment, education, counseling, and therapeutic feeding [7]. Despite these comprehensive strategies, the dual burden of HIV and malnutrition remains prevalent, jeopardizing the national efforts in reaching elimination targets for HIV/AIDS [15].

Evidence on adherence to these nutritional guidelines at the CTC level in Tanzania is not available. The lack of data on guideline adherence and the factors influencing it continues to affect the quality of care and treatment outcomes for PLHIV, slowing progress towards effective HIV management and elimination goals. This study therefore, aimed to evaluate the adherence to the National Guidelines for Nutrition Care and Support and to identify the facilitators and challenges associated with adherence in CTC clinics.

## Materials and methods

### Study design and area

The study was conducted across Care and Treatment Centers (CTCs) in Kinondoni, Temeke, Ilala, Ubungo, and Kigamboni districts of Dar es Salaam, representing urban and suburban

areas [14]. This evaluation used the explanatory sequential mixed-method study, which had two phases which spanned between 29th April and 27th May 2024. A quantitative phase focused on level of adherence based on the observation of the researcher and patient feedback, whose results were used to build on the second phase of qualitative phenomenological design, targeting experiences of health care workers.

## Study population and sampling

All health facility in-charges of the respective CTCs which are public health facilities, and all PLHIV who attended clinics and provided informed consent were included in the study. Healthcare providers were also included if they were employed at one of selected health CTC clinics within the five districts of Dar es salaam, had at least one year experience working in the CTC, and were directly involved in patient care as recommended by the Tanzania Food and Nutrition Centre (TFNC) evaluation questionnaire guideline [7]. Clients experiencing acute medical distress at the time of data collection were excluded to ensure safety and reliability of the data.

For the quantitative phase, a total of 485 PLHIV were approached for the study. Of them, 7 (1.46%) refused to participate due to time constraints, lack of interest and reluctance to share their experience through an interview. Sample size was calculated using formula from Kish and Leslie (1965) with prevalence of PLHIV's adherence set at 50%, and margin of error set at 5%, and accounting for a 20% non-response rate. A multi-stage sampling method was employed to select 25 public health facilities with CTC clinics were selected using a multi-stage sampling method. From each of the facilities 19 participants were conveniently selected. Patients and treatment supporters who attended CTC clinics, health facility in-charges, doctors, nurses, and nutritionists who consented to participate were included. The exclusion criterion was clients experiencing acute medical distress at the time of data collection. For the qualitative phase, healthcare providers were purposively approached and requested to participate after they have consented. The data saturation was reached with 13 participants whose ages ranged from 24 to 51 years, majority being male. Of them, five were clinical officers, four nurses, three nutritionists, and two medical doctors. They had one to 18 years of experience in working at CTCs.

## Variables and measurements

Adherence level was the dependent variable, assessed using the Minimum Nutrition Package (MNP) per National Nutritional guideline. It contains a total of nine recommendations, each carrying equal weight. Adherence was measured using a binary scoring system, with scores of 6 or more indicating adherence and scores below 5 indicating non-adherence, based on the Index Measuring Adherence to Complementary Feeding Guidelines [16,17]. The Minimum Nutrition Package (MNP) provides guidance for healthcare workers on the care of PLHIV and is categorized into four major components (Fig 1): (i) Nutrition Assessment (Conduct anthropometric assessments, Conduct clinical assessments, Perform biochemical assessments, Conduct dietary assessments,) (ii) Education and Counseling (Counsel about nutrition and HIV, Explain the side effects of medications and dietary management, Assessment on food and water safety and hygiene) (iii) Therapeutic/Supplementary Feeding (Provide micronutrient supplements when necessary), and (iv) Referral/Follow-Up Services, (Schedule patients for follow-up care to monitor progress and adherence) [7]. Independent variables including healthcare provider characteristics, health facility characteristics, resources and tools, training and supervision factors and patient characteristics, adopted from the Tanzania Demographic Health Survey (TDHS) [18], Tanzania Food and Nutrition Center (TFNC) [7], and Evaluation of facility level on NACS guide tool [19].

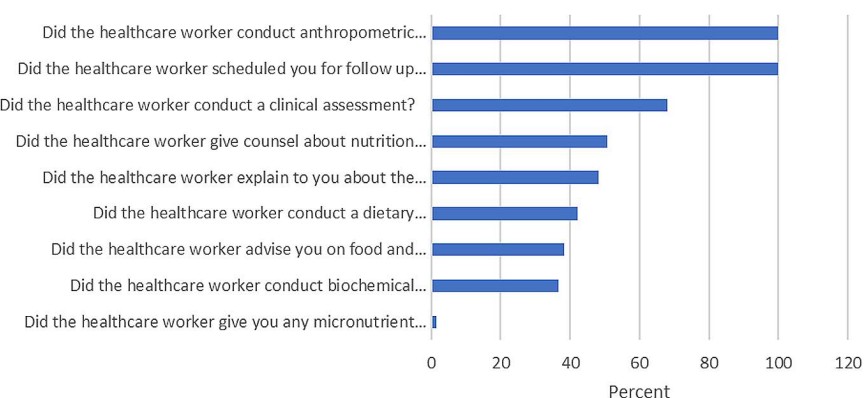

**Fig 1. Adherence to specific aspects of nutritional care guidelines.**

## Data collection and quality assurance

A total of three research assistants were recruited to assist with data collection. Of them, two medical doctors and one clinical officer. They were oriented on the research proposal and trained on its research objectives and ethical consideration. They underwent a comprehensive three-day training program that focused on interview techniques, questionnaire content, data collection ethics, a review of nutritional guidelines for PLHIV.

A structured questionnaire (S1 Text) was used in the quantitative phase. It was developed in English and translated to Kiswahili, a common language used by all residents in Dar es salaam. The Kiswahili questionnaire was uploaded on Kobo Collect, tested on a pilot population at Tiptop and Toagoma dispensaries which were not featured in the main data collection sites. The reliability was tested with Cronbach's Alpha level of 0.71. The collected data were cleaned, processed, and securely stored in cloud storage. To ensure data quality, each day, data was inspected for inconsistency and mitigated further on the following day. Other data quality issues, such as missing data and incomplete responses, were mitigated through participant follow-ups to fill the gaps. For the qualitative study, in-depth interviews were conducted face-to-face using tested semi structured interview guide (S1 Text). Recorded audio and notes were taken on the non- verbal expressions during the interview. Each interview was done in a private room inside the CTC facility to ensure privacy and quality for the recordings and took an average duration of 25 minutes.

## Data management and analysis

For the quantitative data, Stata version 18 (Stata Corp, College Station, TX, USA) was used for analysis. Descriptive statistics were conducted to estimate the adherence to nutritional guidance and characterizing study participants. Regression analyses were used to identify the significant determinants influencing adherence while adjusting for potential confounders. This started with bivariate analysis to determine individual association with adherence to the guidelines. Variables with association that had *p < 0.2* were thereafter subjected to a multiple logistic regression analysis to determine an independent association after controlling for confounders and covariates. A threshold p-value of less than 0.05 was set for statistical significance.

For the qualitative data, analyses began with data familiarization, where audio recordings were transcribed verbatim and then translated from Swahili to English. A code book was developed to identify domains and generate initial codes with one data coder responsible

for coding the data. NVivo software was used to manage and analyze the data, facilitating cross-referencing of the 13 transcripts and organizing codes. Each code was thoroughly analyzed to capture the context and narrative of interviewees, providing a deep understanding of the facilitators and barriers related to adherence.

### Ethical consideration

Permissions were granted by the health facilities involved. Ethical clearance was secured from the MUHAS ethical review committee (Reference No. DA.282/298/01.C/2156). All participants provided written informed consent, which detailed the study's objectives, as well as information on data privacy and confidentiality. Participants who did not receive services in line with the standard guidelines were counseled to address the discrepancies.

## Results

### Quantitative findings

**Socio-demographic characteristics of participants.**  A total of 478 participants were recruited for this study, with 351 (73.43%) being female. The mean age of the participants was 42 years (SD: 12.06). Most participants, 158 (33.1%), were aged 45–54 years. Additionally, 278 (58.16%) of the participants resided in urban areas, and 154 (32.22%) were married. A significant portion, 319 (66.74%), had a primary education, and over the past 12 months, 393 (82.22%) reported having an income source (Table 1).

**Characteristics of selected CTC clinics.**  The study included 25 Care and Treatment Clinics (CTCs) in Dar es Salaam. Among these, dispensaries were 15 (60%), with a mean volume of 2322 clients (SD: 2097.3). Additionally, 19 (76%) of the clinics had at least minimum required number of staffs allocated. However, 16 (64%) did not have staff with necessary training. Furthermore, 15 (60%) of the facilities did not receive any support supervision, and among those received supervision 15 (60%) reported that did not receive feedback. Twenty-four clinics (96%) had an assigned weighing scale, and 23 (92%) had an assigned stadiometer. Moreover, 17 (68%) did not have a MUAC tape, Table 2.

**Adherence of guideline in selected CTCs.**  Out of the 478 participants, 93 (19.46%) PLHIV received treatment which has fully adhere to the national nutrition guidelines. While follow-up care and anthropometric assessments were universally implemented, the provision of micronutrient supplements was significantly lacking, with only 1% of participants receiving them (Fig 1).

Adherence rates to guidelines varied across different types of healthcare facilities. Referral hospitals had the highest adherence rate at 40%, indicating a strong compliance with the guidelines. Hospitals and health centers both showed moderate adherence rates, each at 25%. Dispensaries had the lowest adherence rate at 10%, suggesting potential challenges in maintaining guideline compliance in these facilities (Fig 2).

**Determinants of adherence to guideline.**  In a multivariate analysis, several factors were identified as influencing adherence to nutritional guidelines among participants. Clients with higher education were significantly more likely to receive care that adhered to the National Guidelines for Nutrition Care and Support, with a 5.08 times higher adherence rate compared to those with no education (AOR = 5.08; 95% CI: 1.30–19.79; p = 0.019). Additionally, clients attending referral hospitals had a notably higher likelihood of receiving care that adhered to the National Guidelines, with an 8.23 times higher adherence rate compared to those attending dispensaries (AOR = 8.23; 95% CI: 1.19–56.59; p = 0.032). Conversely, participants who were attended by nurses were less likely to receive care that adhered to the National Guideline for Nutrition Care and Support compared to those attended by doctors, with an

**Table 1. Socio-demographic characteristics of participants (n = 478).**

| Variable | Category | Frequency | Percentage (%) |
|---|---|---|---|
| **Gender** | Female | 351 | 73.43 |
| | Male | 127 | 26.57 |
| **Age** | 15–24 years | 27 | 5.65 |
| | 25–34 years | 87 | 18.20 |
| | 35–44 years | 141 | 29.50 |
| | 45–54 years | 158 | 33.05 |
| | 55–64 years | 50 | 10.48 |
| | 65+ years | 15 | 3.14 |
| **Residence** | Suburban | 200 | 41.84 |
| | Urban | 278 | 58.16 |
| **Marital Status** | Widowed | 61 | 12.76 |
| | Married | 154 | 32.22 |
| | Living Together | 55 | 11.51 |
| | Divorced/Separated | 105 | 21.97 |
| | Never Married | 103 | 21.55 |
| **Income Source** | No | 85 | 17.78 |
| | Yes | 393 | 82.22 |
| **Education Level** | Higher Education | 19 | 3.97 |
| | Secondary Education | 89 | 18.62 |
| | Primary Education | 319 | 66.74 |
| | No Education | 51 | 10.67 |
| **Level of Facility** | Referral Hospital | 58 | 12.13 |
| | Hospital | 40 | 8.37 |
| | Health Center | 124 | 25.94 |
| | Dispensary | 256 | 53.56 |
| **Profession (Cadre)** | Medical Doctor | 185 | 38.7 |
| | Clinical Officer | 196 | 41 |
| | Nurse | 97 | 20.29 |
| **Ever done CD4 test** | No | 35 | 7.43 |
| | Yes | 443 | 92.68 |

AOR of 0.34 (95% CI: 0.70–4.42; p = 0.038). Adequate staffing was associated with a lower likelihood of receiving care that adhered to the National Guideline for Nutrition Care and Support compared to having inadequate staffing, with an AOR of 0.31 (95% CI: 0.13–0.76; p = 0.010). Urban residents were also less likely to receive care that adhered to the Nutrition guideline compared to those in sub-urban areas (AOR = 0.47; 95% CI: 0.27–0.82; p = 0.009), Table 3.

## Qualitative findings

Participants included clinical officers, nurses, nutritionists, and medical doctors with 1 to 18 years of experience in Care and Treatment Clinics (CTCs) within the selected dispensaries, health centers, and hospitals. Their ages ranged from 24 to 51 years, majority being male (Table 4).

Facilitators to adhering to guideline were training and competencies, organizational support, and patient education. Barriers included resource limitations, staff turnover, and documentation challenges, Table 5.

**Table 2.** Characteristics of selected CTC clinics in Dar es Salaam.

| Variable | Category | Frequency | Percentage (%) |
|---|---|---|---|
| **Type of Health Facility** | Referral Hospital | 3 | 12 |
| | Hospital | 2 | 8 |
| | Health Center | 5 | 20 |
| | Dispensary | 15 | 60 |
| **Number of Staff Allocated** | Inadequate | 6 | 24 |
| | Adequate | 19 | 76 |
| **With Trained Staff** | No | 16 | 64 |
| | Yes | 9 | 36 |
| **Received Support Supervision** | No | 15 | 60 |
| | Yes | 10 | 40 |
| **Received Feedback After Supervision** | Don't Know | 2 | 8 |
| | No | 15 | 60 |
| | Yes | 8 | 32 |
| **Weighing Scale Available** | Yes, Assigned | 24 | 96 |
| | Yes, Shared with other clinics | 1 | 4 |
| | No | 0 | 0 |
| **Stadiometer Available** | No | 1 | 4 |
| | Yes, Assigned | 23 | 92 |
| | Yes, Shared with other clinics | 1 | 4 |
| **MUAC Available** | No | 17 | 68 |
| | Yes, Assigned | 7 | 28 |
| | Yes, Shared with other clinics | 1 | 4 |

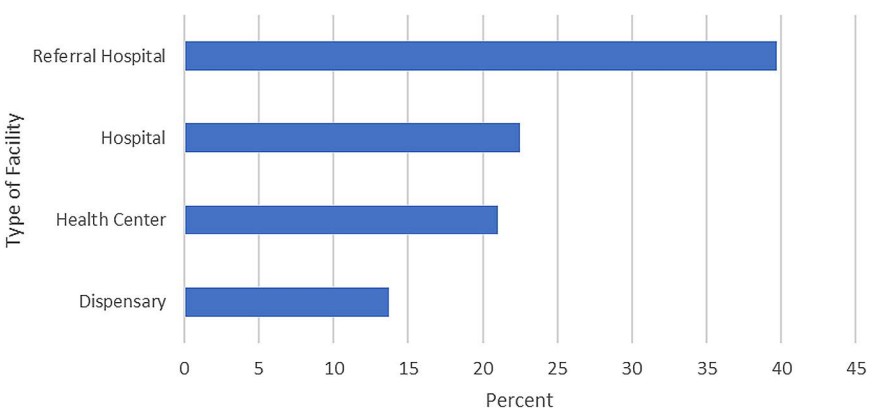

**Fig 2. Adherence to the level of the facilities.**

**Training and competencies.** In this study, healthcare providers understand the importance of adhering to national nutritional guidelines for PLHIV. Their knowledge and abilities have improved greatly because of the training programs offered by different organizations. *"I have attended around ten to fifteen nutrition training sessions related to the national nutrition guidelines Training has significantly improved my skills, they have significantly improved my competency, especially those from MUHAS we understand many things now...* (IDI participant 13)

Table 3. Multivariate logistic regression analysis of factors associated with adherence.

| Variable and Categories | Total (n) | COR (95% CI) | P-Value | AOR (95% CI) | P-Value |
|---|---|---|---|---|---|
| **Age Category** | | | | | |
| **15–24** | 27 | 1 | | Ref | |
| **25–34** | 87 | 0.78 (0.27–2.26) | 0.66 | 1.30 (0.38–0.43) | 0.671 |
| **35–44** | 141 | 0.68 (0.24–1.87) | 0.45 | 0.99 (0.29–3.39) | 0.992 |
| **45–54** | 158 | 0.95 (0.35–2.56) | 0.935 | 1.45 (0.43–4.93) | 0.545 |
| **55–64** | 50 | 0.87 (0.27–2.74) | 0.819 | 1.14 (0.29–4.53) | 0.845 |
| **65+** | 15 | 1.2 (0.29–5.48) | 0.746 | 0.74 (0.13–4.19) | 0.740 |
| **Gender** | | | | | |
| **Female** | 351 | 1 | | Ref | |
| **Male** | 127 | 0.95 (0.56–1.59) | 0.853 | 0.82 (0.45–1.50) | 0.539 |
| **Education Level** | | | | | |
| **No education** | 51 | 1 | | Ref | |
| **Primary** | 319 | 1.10 (0.51–2.3) | 0.803 | 1.42 (0.59–3.40) | 0.423 |
| **Secondary** | 89 | 1.02 (0.41–2.51) | 0.961 | 1.40 (0.50–3.89) | 0.511 |
| **Higher** | 19 | 2.72 (0.83–8.83) | 0.096 | 5.08 (1.30–19.79) | 0.019 |
| **Marital Status** | | | | | |
| **Divorced** | 105 | 1 | | Ref | |
| **Living together** | 55 | 1.5 (0.63–3.53) | 0.354 | 2.03 (0.79–5.21) | 0.141 |
| **Married** | 154 | 1.45 (0.73–2.85) | 0.280 | 1.91 (0.90–4.04) | 0.088 |
| **Never Married** | 103 | 1.725 (0.84–3.53) | 0.136 | 2.03 (0.88–4.64) | 0.093 |
| **Widowed** | 61 | 1.78 (0.79–4.01) | 0.160 | 1.74 (0.70–4.32) | 0.227 |
| **Residence area** | | | | | |
| **Sub-urban** | 200 | 1 | | Ref | |
| **Urban** | 278 | 0.61 (0.38–0.96) | 0.034 | 0.47 (0.27–0.82) | 0.009 |
| **Source income** | | | | | |
| **No** | 85 | 1 | | Ref | |
| **Yes** | 393 | 0.74 (0.422–1.30) | 0.297 | 0.84 (0.42–1.67) | 0.628 |
| **Type of Facility** | | | | | |
| **Dispensary** | 256 | 1 | | Ref | |
| **Health Center** | 124 | 1.67 (0.95–2.93) | 0.071 | 0.88 (043–1.79) | 0.737 |
| **Hospital** | 40 | 1.83 (0.80–4.17) | 0.149 | 1.13 (0.37–3.42) | 0.828 |
| **Referral Hospital** | 58 | 4.14 (2.19–7.83) | 0.000 | 8.23 (1.19–56.59) | 0.032 |
| **Provider Type** | | | | | |
| **Clinical Officer** | 196 | 1 | | Ref | |
| **Nurse** | 97 | 0.33 (1.13–0.84) | 0.020 | 0.34 (0.70–4.42) | 0.038 |
| **Doctor** | 185 | 2.24 (1.37–3.66) | 0.001 | 3.77 (1.72–8.27) | 0.001 |
| **Staff Level** | | | | | |
| **Inadequate** | 6 | 1 | | Ref | |
| **Adequate** | 19 | 0.82 (0.47–1.44) | 0.500 | 0.31 (0.13–0.76) | 0.010 |
| **Trained Staff** | | | | | |
| **No** | 16 | 1 | | Ref | |
| **Yes** | 9 | 2.30 (1.33–3.96) | 0.003 | 0.23 (0.04–1.34) | 0.103 |
| **Supervision Status** | | | | | |
| **No** | 15 | 1 | | Ref | |
| **Yes** | 10 | 2.09 (1.32–3.30) | 0.002 | 0.69 (0.34–1.41) | 0.319 |
| **Tool (MUAC)** | | | | | |
| **No** | 17 | 1 | | | |
| **Yes, assigned** | 7 | 2.59 (1.61–4.18) | 0.000 | 1.70 (0.955–3.04) | 0.071 |
| **Yes, shared** | 1 | 1.07 (0.30–3.84) | 0.908 | | |

**Table 4. Socio-demographic characteristics of participants (n = 13).**

| Participant code | Type of Facility | Gender | Age (Yrs.) | Profession | Experience working at CTC (Yrs.) |
|---|---|---|---|---|---|
| 1 | Dispensary | F | 28 | Clinical Officer | 3 |
| 2 | Health Center | M | 36 | Clinical Officer | 3 |
| 3 | Dispensary | M | 27 | Nurse | 1 |
| 4 | Health Center | M | 38 | Clinical Officer | 2 |
| 5 | Hospital | M | 34 | Nurse | 4 |
| 6 | Hospital | F | 26 | Nutritionist | 1 |
| 7 | Health Center | M | 35 | Medical Doctor | 1 |
| 8 | Health Center | F | 24 | Nutritionist | 1 |
| 9 | Dispensary | F | 51 | Nurse | 18 |
| 10 | Dispensary | F | 40 | Nurse | 10 |
| 11 | Dispensary | M | 28 | Clinical Officer | 2 |
| 12 | Hospital | M | 38 | Medical Doctor | 3 |
| 13 | Hospital | F | 51 | Nutritionist | 16 |

**Table 5. Facilitators and barriers in adhering to the guideline.**

| Categories | Subcategories | Codes |
|---|---|---|
| 1. Facilitators to Adhering to Guidelines | Training and Competencies | Received training programs; Competencies; skills; knowledge; guidelines awareness. |
| | Organizational Support | Leadership support; interdepartmental coordination |
| | Patient Education | Regular educational sessions; Integrated health education programs. |
| 2. Barriers to Adhering to Guidelines | Resource Limitations | Insufficient funding; Lack of nutritional supplements; patient economic status |
| | Staff turnover | High staff turnover; time constraints; workload; Inadequate training for new staff. |
| | Documentation Challenges | Poor documentation by healthcare providers; Lack of nutritional indicator. |

**Organizational support and guideline.** Participants pointed that effective leadership has shown a crucial role in ensuring adherence to nutritional guidelines, particularly in the context of managing HIV. It involves not just supervision but also fostering a supportive and motivating environment for staff to consistently meet established standards. *"The specific ways our leaders use involve internal supervision, which is conducted to monitor the clients served throughout the week. This is because there is something called 'backup.' When this backup is sent, it goes primarily to the Medical Officer In charge, who reviews it since they are the leaders, we have here..."* (IDI participant 3)

This proactive leadership ensures that staff are well-informed about guidelines and receive ongoing support and oversight, promoting a culture of continuous improvement. *"Our leaders remind us every day to follow the guidelines. They provide us with the guidelines, which we read regularly. We also have weekly meetings where they guide us on following the guidelines......They help by providing tools designed to guide us, which they regularly monitor these tools to ensure we are following the guidelines. They check and review the tools, which helps remind us of anything we might have forgotten."* (IDI participant 7)

**Patient education.** Participant explained that Regular educational sessions are essential for enhancing patients' understanding of nutrition, especially for those with HIV. These sessions offer ongoing, organized opportunities to learn about proper nutrition and its significance in managing overall health.

*"We usually provide education every morning before starting services at 7:30 am. The education we provide covers many topics, including nutrition, because the medication they*

*take requires food. Without sufficient education, people cannot understand this, so we emphasize the importance of food, the importance of leafy vegetables, lifestyle nutrition, and which foods are not good for health."* (IDI participant 9)

**Resource limitation.**  Nutritional supplements have been inconsistent, often reliant on fluctuating external funds, this leads to period of unavailability of essential nutritional supplements, which it negatively impacts the health and nutrition of PLHIV. *"Unfortunately, we're facing challenges. We lack therapeutic and supplemental foods, which hinders our progress. A client comes in with needs, really needs nutritional supplements. So, just talking to them and telling them to do one, two, three things set us back. It's frustrating; there's no perfect way to express this".* (IDI participant 8)

**Staff turnover.**  Participant pointed out staff turnover in healthcare, especially in complex care settings like HIV treatment, compromises patient care by disrupting continuity and overburdening remaining staff. *"When you talk about the reduction in healthcare providers going on leave, it significantly affects service delivery because, for example, here in this facility, we have very few staff members. So, when one staff member leaves, the workload becomes much heavier, especially in serving our clients…. Therefore, staff turnover impacts our adherence to guidelines because you won't be able to do everything that is required for the patient."* (IDI participant 3)

**Documentation challenges.**  Poor documentation and the absence of clear nutritional care standards are critical gaps compromising healthcare quality. Incomplete records and lack of specific indicators hinder effective patient care and outcomes. *"Those of us working in the CTC have indicators, so our major task is to engage with those indicators…... We concentrate on working with these indicators. For example, for viral load suppression, we implement all mechanisms to ensure it increases………. However, for nutrition, we don't have such an indicator, which is a challenge..."*(IDI participant 4)

## Discussion

Only nineteen percent of PLHIV received care which fully adhered to the National Nutrition Guidelines, a rate significantly lower than other countries in region like Ethiopia where adherence ranges from 36.3% to 41% [20]. The universal implementation of follow-up care and anthropometric assessments reflects a commitment to some aspects of nutritional care, but highlights insufficiencies in holistic patient management, echoing WHO recommendations for comprehensive care [21,22]. Only one percent of participants received essential micronutrient supplements, indicative of systemic incapacities including supply chain failures. This deficiency has significant implications for immune function and health in HIV-infected individuals, underscoring a gap that is starkly highlighted by WHO's emphasis on the importance of these nutrients [23]. Clinical nutritional assessments were performed for sixty percent of participants, suggesting moderate adherence. However, less than half received dietary assessments, side effects monitoring, hygiene assessments, and nutrition counseling, pointing to significant gaps in nutritional management. These findings are consistent with higher rates of dietary assessments and nutrition counseling reported in studies from Kenya and Uganda, suggesting a regional disparity in adherence practices [24,25].

Educational level was a prominent determinant of adherence, with individuals having higher education showing significantly better compliance, underlining the importance of health literacy in guideline adherence. This finding is supported by a study in Ghana, showing that education correlates strongly with better nutritional outcomes [26]. The type of healthcare facility also critically influenced adherence, with referral hospitals achieving the highest rates (40%) due to better resources and specialized staffing, contrasting sharply with

dispensaries that showed only 10% adherence, highlighting the resource disparity across facility levels. This pattern is echoed in Malawi, where private health facilities noted higher adherence rates [27]. Provider type was another key factor, with doctors achieving higher adherence in their care compared to nurses, suggesting that provider training and qualifications play significant roles in guideline adherence. Resource availability within healthcare facilities also had a substantial impact, with well-resourced facilities demonstrating higher adherence, similar to findings from Mozambique where better-equipped facilities saw more frequent patient visits [28].

Consistent training was reported to be a facilitator linked to improved healthcare delivery, similar to findings in Tanzania that showed significant enhancements in health workers' nutrition knowledge post-training [29]. Strong organizational leadership and robust feedback mechanisms, was identified as another facilitator, enhancing adherence by enabling prompt corrections to deviations from protocols, echoing a study in Mozambique that highlighted the impact of healthcare service availability on care-seeking behaviors. Interdepartmental coordination was essential for integrated patient care, especially for HIV patients requiring multifaceted approaches, aligning with the Mozambican study's emphasis on facility type and resource levels influencing healthcare decisions [28]. Patient education was pivotal in improving health outcomes, supported by findings from Nigeria showing higher adherence rates among patients educated in private health facilities [30].

Conversely, significant barriers included resource limitations, particularly nutritional supplements, which directly affected care quality, a challenge also noted in Mozambique [28]. Staffing turnover and shortages posed substantial barriers, increasing workloads and potentially leading to caregiver burnout, necessitating strategic workforce planning to maintain care continuity, a strategy supported by findings from Uganda [31,32]. Documentation challenges impaired nutritional care quality, where poor practices hindered effective intervention monitoring, a situation improved by standardized documentation practices advocating for electronic health records in the country [33]. Economic constraints also critically impacted patients' adherence to nutritional guidelines, with financial hardships forcing compromises in dietary quality, a dynamic observed in a study on the interplay between economic conditions and healthcare outcomes [34]. Integrated support services providing both nutritional and economic assistance were deemed vital for enabling patients to adhere to dietary recommendations effectively [35].

The study had certain limitations. The external validity of findings were limited by the unique healthcare infrastructure and socio-economic context of Dar es Salaam; which may not be accurate representative of the less developed regions. Additionally, there was potential of response bias, as data from patients and healthcare providers were partially self-reported. However, the inclusion of an observational component helped mitigate this bias to some extent.

## Conclusion

Our study found a significant gap in adherence to national nutritional guidelines among HIV patients, with only 19% receiving care that fully adhered to the guideline, substantially lower than findings from similar studies. While follow-up care and anthropometric assessments were robustly implemented, a severe shortfall in essential micronutrient supplements suggests systemic failures, potentially involving supply chain and funding issues. Variations in adherence across different healthcare facilities highlights the importance of resource distribution, staff training, and facility capabilities. Socio-demographic factors such as education levels and geographical location also significantly influenced adherence outcomes. Enhancing training for healthcare providers through continuous, independent programs to maintain high

standards of care, strengthening resource allocation to lower-tier health facilities, and expanding patient education to improve adherence. Additionally, addressing systemic barriers such as shortage of essential supplements and improving documentation practices are for effective patient management and improved health outcomes.

## Supporting information

**S1 Text. Questionnaire, checklist and guide.** (PDF)

## Acknowledgments

The authors wish to acknowledge dedicated academic staff at the School of Public Health, and the Department of Development Studies at MUHAS, the Regional AIDS Control Coordinator office and the CTC clinics in Dar es salaam whose organized efforts, professional guidance, and unwavering support created a research-friendly environment that was crucial to the conceptualization and implementation of this study.

## Author contributions

**Conceptualization:** Felistar Mwakasungura, Rebecca Mkumbwa, Bruno Sunguya.

**Data curation:** Felistar Mwakasungura.

**Formal analysis:** Felistar Mwakasungura.

**Funding acquisition:** Felistar Mwakasungura.

**Investigation:** Felistar Mwakasungura.

**Methodology:** Felistar Mwakasungura, Bruno Sunguya.

**Project administration:** Felistar Mwakasungura.

**Supervision:** Rebecca Mkumbwa, Bruno Sunguya.

**Validation:** Bruno Sunguya.

**Writing – original draft:** Felistar Mwakasungura.

**Writing – review & editing:** Rebecca Mkumbwa, Bruno Sunguya.

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
