## [Decision Letter · Decision Letter 0]

11 Oct 2024

PGPH-D-24-01851

Facilitators and barriers for healthcare workers' adherence to the national nutritional guidelines for people living with HIV in Dar-es-Salaam: A mixed-method study

Dear Dr. Mkumbwa,

Thank you for submitting your manuscript to PLOS Global Public Health. After careful consideration, we feel that it has merit but does not fully meet PLOS Global Public Health’s publication criteria as it currently stands. Therefore, we invite you to submit a revised version of the manuscript that addresses the points raised during the review process.

The manuscript has been evaluated by two reviewers, and their comments are available below and in the attached document.

The reviewers have raised a number of concerns and request clarification and additional detail regarding the methodological aspects of the study.Could you please carefully revise the manuscript to address all comments raised?

We look forward to receiving your revised manuscript.

Kind regards,

Steve Zimmerman, PhD

PLOS Staff Editor

Journal Requirements:

Additional Editor Comments (if provided):

Reviewers' comments:

Reviewer's Responses to Questions

**Comments to the Author**

1. Does this manuscript meet PLOS Global Public Health’s publication criteria ? Is the manuscript technically sound, and do the data support the conclusions? The manuscript must describe methodologically and ethically rigorous research with conclusions that are appropriately drawn based on the data presented.

Reviewer #1: Yes

Reviewer #2: Yes

2. Has the statistical analysis been performed appropriately and rigorously?

Reviewer #1: Yes

Reviewer #2: Yes

3. Have the authors made all data underlying the findings in their manuscript fully available (please refer to the Data Availability Statement at the start of the manuscript PDF file)?

Reviewer #1: Yes

Reviewer #2: Yes

4. Is the manuscript presented in an intelligible fashion and written in standard English?

Reviewer #1: Yes

Reviewer #2: Yes

5. Review Comments to the Author

Reviewer #1: Authors need to improve the methods section which lacks the most important elents such as how anthropometric measurements were taken; mention who collected the data; was training given to data collectors and explicitly mention how data quality was assured. secondly some improvements in the language is also necessary.

Reviewer #2: Line 2: Could you please update the data to reflect HIV statistics for 2023, rather than 2022, in the second line?

Line 52: You mention progress towards the 95-95-95 targets. Could you elaborate on the current status of the 95-95-95 target in Tanzania? Please include this information in your introduction, and discuss its implications on malnutrition.

Lines 63-65: In the "Study Design" section, the statistics provided here can be removed and integrated into the introduction.

Lines 71-72: Can you provide more details regarding why 7 participants (1.46%) refused to participate in the study?

Regarding your semi-structured questionnaire for qualitative data:

- Was it developed specifically for this study, or adopted from previous studies?

- If it was developed for this study, please include the interview guide as a supplementary file.

- If it was adopted from previous studies, please provide references.

Did you follow all the necessary steps for reporting qualitative studies? If so, please submit the Consolidated Criteria for Reporting Qualitative Research (CORE-Q) as supplementary files.

Line 181: Is this section presenting quantitative or qualitative results? If it is part of the qualitative findings, could you provide a table detailing your participants' experiences and professional backgrounds?

Please ensure that the study’s potential limitations are clearly highlighted.

Finally, the conclusion is too long. Kindly revise it to deliver a clearer, more concise message for the reader.

6. PLOS authors have the option to publish the peer review history of their article (what does this mean? ). If published, this will include your full peer review and any attached files.

**Do you want your identity to be public for this peer review?** For information about this choice, including consent withdrawal, please see our Privacy Policy .

Reviewer #1: No

Reviewer #2: **Yes: ** Pierre Gashema

---

## [Decision Letter · Decision Letter 1]

30 Jan 2025

Facilitators and barriers for healthcare workers' adherence to the national nutritional guidelines for people living with HIV in Dar-es-Salaam: A mixed-method study

PGPH-D-24-01851R1

Dear Ms Mkumbwa,

We are pleased to inform you that your manuscript 'Facilitators and barriers for healthcare workers' adherence to the national nutritional guidelines for people living with HIV in Dar-es-Salaam: A mixed-method study' has been provisionally accepted for publication in PLOS Global Public Health.

Best regards,

Julia Robinson

Executive Editor

Reviewer Comments (if any, and for reference):

Reviewer's Responses to Questions

**Comments to the Author**

1. If the authors have adequately addressed your comments raised in a previous round of review and you feel that this manuscript is now acceptable for publication, you may indicate that here to bypass the “Comments to the Author” section, enter your conflict of interest statement in the “Confidential to Editor” section, and submit your "Accept" recommendation.

Reviewer #2: All comments have been addressed

2. Does this manuscript meet PLOS Global Public Health’s publication criteria ? Is the manuscript technically sound, and do the data support the conclusions? The manuscript must describe methodologically and ethically rigorous research with conclusions that are appropriately drawn based on the data presented.

Reviewer #2: Yes

3. Has the statistical analysis been performed appropriately and rigorously?

Reviewer #2: Yes

4. Have the authors made all data underlying the findings in their manuscript fully available (please refer to the Data Availability Statement at the start of the manuscript PDF file)?

Reviewer #2: Yes

5. Is the manuscript presented in an intelligible fashion and written in standard English?

Reviewer #2: Yes

6. Review Comments to the Author

Reviewer #2: (No Response)

7. PLOS authors have the option to publish the peer review history of their article (what does this mean? ). If published, this will include your full peer review and any attached files.

**Do you want your identity to be public for this peer review?** For information about this choice, including consent withdrawal, please see our Privacy Policy .

Reviewer #2: **Yes: ** Pierre Gashema
